# The Evaluation of Significance of Uncoupling Protein Genes *UCP1*, *UCP2*, *UCP3*, *UCP4*, *UCP5,* and *UCP6* in Human Adaptation to Cold Climates

**DOI:** 10.3390/biology14050454

**Published:** 2025-04-23

**Authors:** Alena A. Nikanorova, Nikolay A. Barashkov, Vera G. Pshennikova, Sergey S. Nakhodkin, Georgii P. Romanov, Aisen V. Solovyev, Sardana A. Fedorova

**Affiliations:** 1Yakut Science Centre of Complex Medical Problems, Yaroslavskogo 6/3, 677000 Yakutsk, Russia; nikanorova.alena@mail.ru (A.A.N.); psennikovavera@mail.ru (V.G.P.); 2M.K. Ammosov North-Eastern Federal University, Kulakovskogo 46, 677013 Yakutsk, Russia; sergnahod@mail.ru (S.S.N.); gpromanov@gmail.com (G.P.R.); nelloann@mail.ru (A.V.S.); sardaanafedorova@mail.ru (S.A.F.)

**Keywords:** uncoupling proteins, *UCP1*, *UCP3*, free triiodothyronine (FT3), non-shivering thermogenesis, shivering thermogenesis, cold climate, adaptation

## Abstract

In this study, we present an analysis of SNP variants of six uncoupling protein genes in the Yakut population living in the coldest region of Siberia (minimum temperature −71.2 °C) in the context of human adaptation to cold climates. Uncoupling protein 1 (UCP1) is a transmembrane protein that may play an important role in thermogenesis; the functional role of other UCPs is clearly not known. Based on our results, we suggest that uncoupling proteins UCP1 and UCP3 are involved in human adaptation to a cold climate by increasing heat production. Thus, understanding the molecular details of the functional role of the UCPs could pave the way for the development of novel therapeutics to curb metabolic diseases such as obesity and diabetes.

## 1. Introduction

Uncoupling proteins (UCPs) are transmembrane proteins that mediate specific metabolite exchange between the cell cytoplasm and mitochondrial matrix [1]. Currently, proteins of this family are represented by six isoforms (UCP1–UCP6), which are present in various tissues [2,3,4]. Among them, UCP1 is the most well-studied isoform, which is mainly expressed in brown adipose tissue (BAT) [5]. In contrast, the second isoform, UCP2, is found in a variety of tissues, including white adipose tissue, skeletal muscle, the heart, and immune cells [6,7,8,9,10,11]. UCP3 is predominantly expressed in skeletal muscle and the heart [12], while UCP4 and UCP5 are more clearly detected in brain neurons [13,14]. UCP6 mRNA was found to be expressed mainly in mouse kidneys [15]. The existence of UCP6 mRNA in humans was predicted in the seminal ducts, testis, seminal vesicles, and prostate gland [16]. A precise physiological function has only been established for UCP1, which is involved in the non-shivering thermogenesis of BAT [17,18], but the functions of other UCPs are still not fully defined.

It is known that thyroid hormones, such as triiodothyronine (T3) and thyroxine (T4), could be involved in the regulation of *UCP1* gene expression during non-shivering thermogenesis of BAT [19,20,21,22]. It has been demonstrated that cold stress in brown adipocytes, through the deiodination of T4 to T3, increases concentrations of active T3, which penetrates the nucleus and forms a complex with its thyroid hormone nuclear receptors (TRs), and then this T3-TR complex binds to thyroid response elements (TRE) on the *UCP1* gene stimulating its expression [19,20,21,22]. The expression of other *UCP* genes, such as *UCP2* and *UCP3*, is regulated by the same mechanism. In rodents, UCP2 and UCP3 expression have been shown to be enhanced by T3 in muscles [23,24]. A direct effect of T3 on UCP2 and UCP3 mRNA expression in skeletal muscle was demonstrated in vitro in human primary cultures [25]. The results of in vivo studies by de Lange et al. [26] showed that T3 regulates UCP3 mRNA levels in mitochondria, and UCP3 could potentially act as a molecular determinant in the regulation of resting metabolic rate by T3. However, the role of thyroid hormones in the regulation of other uncoupling protein genes, *UCP4*, *UCP5*, and *UCP6*, is currently unknown.

On the other hand, it is known that thyroid hormones play an important role in regulating the basal metabolic rate [27]. Previous studies have shown that the basal metabolic rates of indigenous peoples in Siberia, such as the Yakuts, Evenks, and Buryats, are higher than predicted values for European populations [28,29,30,31]. This is thought to be due to thyroid hormones [32]. It is assumed that for the survival of the organism during cold stress, the dynamics of thyroid hormones may shift towards a type 2 allostasis, characterized by an increase in circulating T3 levels, leading to a higher basal metabolic rate to maintain thermogenic mechanisms [33,34]. Currently, it is proposed to use the parameters of thyroid homeostasis, SPINA (SPINA-GD and SPINA-GT), for the differential diagnosis of an allostasis reaction, including cold stress [35]. SPINA-GD indicates the total activity of peripheral deiodinases, while SPINA-GT shows the maximum production of T4 by the thyroid gland per unit of time [36]. Previously, changes in the homeostasis of the pituitary–thyroid axis (type 2 allostasis) at low atmospheric temperatures (−47 °C to −11 °C) were detected in 70% of the individuals living in the extremely cold climatic conditions of Eastern Siberia [37].

Additionally, indigenous northern populations in Eurasia and North America tend to have more compact physiques, with relatively longer torsos and shorter limbs, as well as a larger body weight compared to their shorter height, i.e., smaller body surface area (BSA) [38,39,40,41,42]. Based on studies of the effects of cold air on humans, common morphological features associated with adaptation to cold have been identified among indigenous people of temperate and polar climates [30,43,44,45]. These morphological features of the physique allow the body to keep warm for a longer time, since at a lower BSA-to-weight ratio, a slower decrease in body temperature is observed during exposure to cold [46,47].

Recently, a number of studies have been conducted aiming to discover genes with adaptive variants for cold climate tolerance. Several studies have identified associations of some variants of *UCP* genes with human adaptation to cold climates. Hancock et al. [48] hypothesized that such polymorphic variants as *UCP1*-rs1800592, *UCP2*-rs659366, and *UCP3*-rs1800849, which are associated with increased expression of these genes, may be involved in human adaptation to cold climates. Based on the analysis of the correlation of allele frequencies in 52 world populations with winter climate variables for polymorphic variants, they concluded that the rs1800592 *UCP1* and rs1800849 *UCP3* variants have a significant correlation with winter climate [48]. Another group of researchers studied allele frequencies of 28 genes potentially associated with adaptation to cold climate in populations of Northern Eurasia [49]. As a result of these studies, a significant association of the rs1800592 polymorphism of the *UCP1* gene with climatic (temperature) and with geographic (latitude and longitude) variables was established, but no signals of directional selection were found [49]. A study focused on identifying cold adaptation traits in ancient hunter-gatherers in Japan (the Jomon people) revealed associations between four SNPs in the *UCP1* gene (rs3113195, rs12502572, rs1800592, rs4956451) and the non-shivering thermogenesis phenotype [50]. It has been hypothesized that East Eurasian hunter-gatherers adapted to cold climates through BAT thermogenesis mediated by *UCP1* [50]. In addition, associations of polymorphic variants of *UCP1* (rs3811787) and *UCP3* (rs1800849) genes with the levels of circulating blood hormones leptin and irisin were previously found in residents of Eastern Siberia [51,52]. Based on the association analysis, a potential role of the *UCP1* gene in the leptin-mediated non-shivering thermogenesis pathway, as well as a role of the *UCP3* gene in shivering thermogenesis, was suggested [51,52].

In this regard, in the present work, for the first time, the search for signs of participation of nine polymorphic variants of *UCP1*, *UCP2*, *UCP3*, *UCP4*, *UCP5,* and *UCP6* genes in human adaptation to cold climate was carried out by evaluating the strength of influence using four different criteria: (1) the presence of associations between polymorphic variants of *UCP* genes and levels of hormones of the pituitary–thyroid axis (thyroid-stimulating hormone—TSH, free triiodothyronine—FT3, and free thyroxine—FT4); (2) the presence of associations between polymorphic variants of *UCP* genes and changes in thyroid homeostasis (SPINA); (3) the presence of associations between polymorphic variants of *UCP* genes and BSA; (4) the presence of signals of directional selection for cold climate for polymorphic variants of *UCP* genes.

## 2. Materials and Methods

### 2.1. Subjects

The sample consisted of 279 individuals (185 females and 94 males), and the mean age of the participants was 19.73 ± 1.99 years. No participants had any health complaints at the time of the study, and they filled out a questionnaire themselves, in which they indicated their sex, ethnicity, age, chronic diseases, and experience of taking antidepressants. All participants provided written informed consent to participate in the study. The study was approved by the local biomedical ethics committee of the Yakutsk Scientific Center for Complex Medical Problems of the Siberian Branch of the Russian Academy of Medical Sciences, Yakutsk, Russia (Protocol No. 45, 12 October 2017).

### 2.2. Anthropometric Parameters

Anthropometric parameters (body weight in kilograms and height in centimeters) were measured in all participants (n = 279) using standardized methods. The body mass index was calculated by dividing body weight by the square of height. For association analysis of polymorphic variants of *UCP* genes with changes in thyroid homeostasis (SPINA) the sample (n = 279) was divided into three groups according to body mass index categories [53]: underweight (≤18.49 kg/m^2^), normal weight (18.5–24.99 kg/m^2^), and overweight/obese (≥25 kg/m^2^) (Appendix A). BSA (n = 279) was calculated according to Haycock’s formula [54].

### 2.3. Hormonal Measurement

Venous blood for the study was collected in the morning after an 8 h fast from 94 participants. Levels of TSH (µU/mL), FT3 (pmol/L), and FT4 (pmol/L) in serum were detected by time-resolved immunofluorescence analysis using DELFIA hTSH Ultra, DELFIA Free Thyroxine, and DELFIA Free Triiodothyronine kits (PerkinElmer Inc., Waltham, MA, USA), respectively. The concentrations of the three hormones in the samples were measured at a wavelength of 450 nm on a VICTOR X5 Multilabel Plate Reader (PerkinElmer Inc., USA). The reference values according to the kit recommendations were TSH 0.63–4.2 µU/mL, FT3 4.6–7.8 pmol/L, and FT4 9.8–16.8 pmol/L. TSH levels in all study participants (n = 94) were within normal limits (2.20 ± 0.83 μU/mL); one individual was found to have elevated levels of FT3 (7.96 pmol/L) and FT4 (17.2 pmol/L), and five individuals showed elevated levels of FT4 (17 pmol/L to 18.8 pmol/L). For further analysis, the sample was normalized, and the males with elevated levels of FT3 and FT4 were excluded (n = 6).

### 2.4. PCR-RFLP Analysis

Genomic DNA was isolated from blood using phenol-chloroform extraction. Genotyping of 9 polymorphisms of *UCP1-6* genes was performed for all participants (n = 279) using the polymerase chain reaction-restriction fragment length polymorphism (PCR-RFLP) assay. Data, including primer sequence, annealing temperature, PCR product size, and restriction enzymes, are presented in Table 1. PCR was performed on a Bio Rad T100 thermocycler (Bio-Rad Laboratories, Inc., Berkeley, CA, USA). Genotype and allele frequencies of 9 polymorphic variants of *UCP1-6* uncoupling protein genes are presented in Appendix A.

### 2.5. SPINA Parameters

Thyroid homeostasis assessment parameters were calculated for 94 individuals using the SPINA Thyr software (version 4.1.0, SPINA Thyr, RRID:SCR_014352, doi 10.5281/zenodo.3596049). The software is based on a mathematical model of thyroid hormone homeostasis. The results of the evaluation studies and the algorithms underlying the mathematical theory were published in several papers [55,56,57]. Reference values according to SPINA Thyr program recommendations were SPINA-GT—1.4–8.7 pmol/s, and SPINA-GD—20–40 nmol/s.

### 2.6. Search for Natural Selection Signals for Polymorphic Variants of UCP Genes

Data on the frequencies of the studied alleles were obtained from the “1000 Genomes Project” open database [58] for the following 31 populations: Esan (Nigeria), Gambians (Gambia), Luhya (Webuye, Kenya), Mende (Sierra Leone), Yoruba (Ibadan, Nigeria), African Caribbean (Barbados), Finns (Finland), Britons (England and Scotland), Iberians (Spain), Tuscans (Italy), Bengalis (Bangladesh), Gujarati Indians (USA), Indian Telugu (UK), Punjabis (Lahore, Pakistan), Sri Lankan Tamils (UK), Chinese Dai (Xishuangbanna, China), Han Chinese (Beijing, China), Han Chinese South (China), Japanese (Tokyo, Japan), Vietnamese (Ho Chi Minh City, Vietnam), Mexicans (United States, Mexico), Puerto Ricans (Puerto Rico), Colombians (Medellin, Colombia), Peruvians (Lima, Peru), Estonians (Estonia), Dutch (Netherlands), indigenous Oceanians (Oceania), Koreans (Republic of Korea), Swedes (Sweden), Qataris (Qatar), Danes (Denmark). Additionally, from the published studies [49,59,60], the data on the allele frequencies of rs1800849 of the *UCP3* gene were obtained for 8 populations: Nivkhi (Russia), Koryak (Russia), Chukchi (Russia), Buryats (Russia), Khanty (Russia), Kets (Russia), Brazilians (Brazil), and Turks (Turkey). The map illustrating allele frequency distribution in populations across North and South America, Eurasia, and Africa was created using Surfer 12.0 software (Golden Software, Golden, CO, USA).

### 2.7. Evidence of Involvement of Polymorphic Variants of UCP Genes in Adaptation to Cold Climate

For the first time, a method utilizing four criteria has been developed to assess the role of polymorphic variants in *UCP* genes in human adaptation to cold environments. The following criteria were applied: (1) the presence of associations of polymorphic variants of *UCP* genes with the levels of hormones of the pituitary–thyroid axis (TSH, FT3 and FT4) (n = 94), (2) the presence of associations of polymorphic variants of *UCP* genes with changes in thyroid homeostasis (SPINA) (n = 61), (3) the presence of associations of polymorphic variants of *UCP* genes with BSA (n = 279), (4) the presence of directional selection natural signals for polymorphic variants of *UCP* genes (n = 279). Each criterion was evaluated based on the strength of its influence, expressed in points (ranging from 0 to 3), according to the significance level of the *p*-value determined by the results of the association analyses (Table 2). The maximum total score for all four criteria was 21 points. To determine the minimum threshold, we used a cut-score percentage method. In this method, we fixed the 25th percentage as the minimum threshold. The study design is presented in the Appendix A).

### 2.8. Statistical Analysis

The results were analyzed using the computer software for statistical data editing, Statistica 13.5 (TIBCO Software Inc., Santa Clara, CA, USA). Values of *p* ≤ 0.05 were considered statistically significant. Quantitative results are reported as the mean ± standard deviation. Comparative analysis between the three genotypes was performed using the Kruskal–Wallis H test. Comparative analysis between two genotypes and alleles was carried out using the Mann–Whitney U test.

## 3. Results

### 3.1. Assessment of the Contribution of Polymorphic Variants of UCP Genes in Human Adaptation to Cold

For the first time, the combined approach for analyzing the contribution of polymorphic variants of *UCP* genes in human adaptation to cold was developed. This approach was based on four criteria that can show signs of involvement of genes and their variants in cold adaptation. For each criterion, statistical analyses were performed for nine polymorphic variants of *UCP1-6* genes (Appendix A). According to the results of these analyses and the evaluation of the strength of the criteria influence on cold adaptation, certain scores (from 0 to 3) were assigned for the nine polymorphisms of *UCP* genes (Table 3). Based on the minimum threshold of six points, two gene variants, *UCP1* (rs3811787) and *UCP3* (rs1800849), were found to be most involved in the process of adaptive thermogenesis, as shown in Table 3. The results indicate the involvement of these variants in human adaptation to a cold climate. The rs3811787 (−412A > C) polymorphism of the *UCP1* gene is located at the − 412 position upstream of the transcription start site, and it is in moderate linkage disequilibrium (r^2^ = 0.53) with the second polymorphism rs1800592 [61]. Perhaps for this reason, the frequencies of these two polymorphisms of the *UCP1* gene (rs1800592 and rs3811787) are similar, and both have positive selection signals for cold climates. The rs1800849 (−55C/T) polymorphism of the *UCP3* gene is situated in the promoter region, and the second polymorphism, rs2075577 (+2546T/C) of this gene is located in exon 3 [62]. We assume that these two polymorphisms are independent of each other, and therefore, for the rs2075577 polymorphism, no associations or natural selection signals similar to those observed for rs1800849 were found.

Some other polymorphic variants of the *UCP* genes scored less than six points, indicating their indirect involvement in human adaptation to cold climates (Table 3). The polymorphic variant rs1010978 of the *UCP5* gene was associated with the first criterion (FT3) and was assigned two points. According to the other criteria, there were no associations found for this variant of the *UCP5* gene. Additionally, the search for natural selection signals for the T rs1010978 allele of the *UCP5* gene revealed that the high frequency of this allele is characteristic of warm climates (83–99%). Another polymorphic variant, rs9526067, of the *UCP6* gene has been shown to be associated with two criteria, TSH and SPINA-GT, so it was awarded five points. The search for natural selection signals for the A rs9526067 allele showed that a high frequency of distribution of this allele is found in the Eastern regions of Eurasia (81–92%), as in the south (92%), and in the north (81%). However, the observed associations between uncoupling protein genes and levels of pituitary–thyroid hormones, as well as thyroid homeostasis (SPINA), indicate the existence of several biochemical processes regulated by thyroid hormones in which the uncoupling proteins UCP5 and UCP6 are involved, but this requires further research.

### 3.2. Effect of Polymorphic Variant rs3811787 of the UCP1 Gene on Human Adaptation to Cold

The scoring results revealed that the rs3811787 polymorphism of the *UCP1* gene received a total of seven points. Carriers of the homozygous TT genotype exhibited significantly higher levels of FT3 (7.26 ± 0.30 pmol/L; *p* = 0.02) compared to carriers of the heterozygous GT genotype (6.98 ± 0.35 pmol/L) and the homozygous GG genotype (6.85 ± 0.33 pmol/L) (Figure 1a). Individuals with the TT genotype showed significantly higher SPINA-GD values (44.56 ± 5.12 nmol/s; *p* = 0.02) than carriers of the GT + GG genotype (41.44 ± 6.08 nmol/s) (Figure 1b). No associations of the rs3811787 polymorphism with BSA were observed (Figure 1c). The T allele of rs3811787 demonstrated a signal of directional selection for cold climate adaptation. Low frequencies of the T allele (rs3811787) are common throughout Africa (20–37%), where warm and hot climates prevail. High frequency distribution is characteristic of the northwestern regions of Europe (86–78%), where the climate is considered temperate, transitioning to the subarctic type of climate (Figure 1d).

### 3.3. Effect of the rs1800849 Polymorphic Variant of the UCP3 Gene on Human Adaptation to Cold

The scoring revealed that the strongest contribution to human adaptation to cold climate was made by the polymorphic variant rs1800849 of the *UCP3* gene (six points). Association analysis showed that the TT genotype of rs1800849 was significantly associated with more elevated blood FT3 levels (6.1 ± 0.29 pmol/L; *p* = 0.05) compared to the opposite CC + CT genotype (5.83 ± 0.41 pmol/L) (Figure 2a). No association with SPINA-GD was detected for rs1800849 (*UCP3*) (Figure 2b). Association analysis showed that the TT genotype of rs1800849 was significantly associated with lower BSA (1.61 ± 0.17 m^2^; *p* = 0.04) compared to the opposite CC + CT genotype (1.66 ± 0.19 m^2^) (Figure 2c). For the T rs1800849 allele, signals of directional selection for cold climate adaptation were detected, where a high frequency of the T rs1800849 allele was detected in regions dominated by temperate and arctic climates (Figure 2d).

## 4. Discussion

Based on the evaluation of the contribution of polymorphic variants of *UCP* genes to human adaptation to cold, it was revealed that two polymorphisms, rs3811787 of the *UCP1* gene and rs1800849 of the *UCP3* gene, showed the strongest influence. According to the functional role of *UCP1* and *UCP3* genes, it can be assumed that their role in cold adaptation may be related to heat generation processes during shivering and non-shivering thermogenesis [1,5,17,18,19,20,21,22,23,24,25,26].

### 4.1. The Role of the rs3811787 of the UCP1 Gene in Non-Shivering Thermogenesis

In the present study, a significant involvement of the rs3811787 polymorphic variant of the *UCP1* gene in cold climate adaptation (seven points) was demonstrated, which may be associated with non-shivering thermogenesis in BAT. It was found that the TT genotype of rs3811787 is associated with higher T3 and SPINA-GD levels, as well as with cold climate. Therefore, it is possible that when exposed to cold, active BAT may shift thyroid hormone homeostasis toward type 2 allostasis, which leads to increased levels of FT3 (elevated values peripheral deiodination—SPINA-GD) and accelerated basal metabolism and increased energy expenditure. In addition, the BSA values tend to be higher in carriers of the TT genotype, which also supports the increased basal metabolism (Figure 1c). In turn, it has already been shown that in winter, people living in Eastern Siberia have increased basal metabolic rates by 5.8–6% [63]. However, the increase in thyroid hormone levels during type 2 allostasis can be mitigated by the polar T3 syndrome, which is characterized by decreased levels of FT3 in winter [64,65]. The detailed mechanism of BAT-mediated regulation of thyroid hormones and basal metabolic rate is presented in Figure 3.

In turn, Fernández-Verdejo R. et al., based on the results of multiple studies on energy expenditure during active BAT, suggested that clinically meaningful weight loss would require maximally activated BAT throughout the day, which is unlikely [66]. The maximum capacity for non-shivering thermogenesis in BAT becomes inversely proportional to mass as we get older, and in mammals weighing more than 10,000 g, there is little or no predicted thermal contribution [67,68]. We assume that active BAT is maintained in adult humans not at all for direct heat generation, but rather to accelerate peripheral deiodination (conversion of T4 to T3), which in turn shifts homeostasis of the pituitary–thyroid axis toward type 2 allostasis, resulting in an increase in basal metabolism that is crucial for supporting the survival mechanisms for survival of an organism exposed to chronic cold stress.

### 4.2. The Role of the rs1800849 Polymorphism of the UCP3 Gene in Shivering Thermogenesis

The current study demonstrates that the rs1800849 polymorphic variant of the UCP3 gene significantly affects human adaptation to cold climates (six points), which may be associated with “uncoupling to survive” in skeletal muscles during shivering thermogenesis. It was previously assumed that in skeletal muscle mitochondria, UCP3 participates in “uncoupling to survive” by creating a proton leak in the respiratory chain, resulting in the conversion of electrical charge into thermal energy at the expense of ATP synthesis [69]. Studies on model animals and in vivo models have confirmed the uncoupling ability of UCP3 in skeletal muscle mitochondria [70,71,72,73,74]. Mice with a *UCP3* gene knockout (UCP3 KO) exhibited enhanced mitochondrial respiration, indicating the protein’s proton transport activity [70,71]. In *UCP1* knockout mice, the loss of thermogenesis in brown adipocytes was compensated by increased shivering thermogenesis in muscles, accompanied by elevated UCP3 expression [72]. Another study on UCP3 KO mice confirmed that UCP3 is a critical mediator of physiological thermogenesis in skeletal muscles [73]. An in vivo study demonstrated that UCP3 overexpression in skeletal muscles reduces the ATP synthesis-to-mitochondrial oxidation ratio, suggesting uncoupling via UCP3 [74].

It is hypothesized that UCP3-dependent “uncoupling to survive” is associated with less prolonged shivering thermogenesis but with a greater thermal contribution [40,75]. Other studies have found that cold-adapted individuals exhibited a lower shivering threshold (short-term act of shivering) [76,77]. In turn, the TT genotype rs1800849 of the *UCP3* gene showed a tendency with less efficient skeletal muscle contractions; such effects may be related to UCP-dependent differences in mitochondrial proton leak/uncoupling, and thus inefficiency of ATP genesis relative to oxygen consumption [78]. These data are consistent with our earlier results, where we found associations of the TT genotype rs1800849 with increased levels of irisin in female from Eastern Siberia, indicating that carriers of the TT genotype, due to increased heat production, use less irisin (less shivering) for UCP3-dependent “uncoupling to survive” processes [52]. The mechanism of UCP3-dependent “uncoupling to survive” during shivering thermogenesis is presented in Figure 4.

In the current study, the TT genotype of rs1800849 was found to be associated with increased levels of FT3 in the blood. These results confirm the involvement of T3 in the regulation of expression of the *UCP3* gene [23,26]. However, UCP3-dependent “uncoupling to survive” has side effects that potentially affect BSA (growth and body weight). In the present study, rs1800849 of the *UCP3* gene was found to be associated with BSA, where carriers of the TT genotype had a lower BSA compared to other genotypes. It is possible that a person with a smaller BSA will keep warm for a longer time. Moreover, this allele rs1800849 (*UCP3*) was associated with reduced weight and height in females from Eastern Siberia [64]. It is hypothesized that in carriers of the rs1800849 T allele, the lower BSA (reduced weight and height) may be due to competition of UCP3-dependent “uncoupling to survive” with hormones such as testosterone, growth hormones, and estrogens that utilize ATP to support anabolic pathways [34,79,80].

## 5. Conclusions

Two polymorphic variants of *UCP1* (rs3811787) and *UCP3* (rs1800849) genes, out of nine analyzed polymorphic variants of uncoupling protein genes (*UCP1*, *UCP2*, *UCP3*, *UCP4*, *UCP5*, and *UCP6*), demonstrated a direct involvement in human adaptation to cold climates. The other seven polymorphic variants of *UCP* genes have scored fewer points, so it is assumed that their contribution to human adaptation to cold is less significant;The results we obtained on the association of the TT genotype of rs3811787 in the *UCP1* gene with increased FT3 levels, and elevated SPINA-GD value, in the absence of association with BSA, indicate that the active form of UCP1 in brown adipocytes may utilize more T3, additionally extracting T3 from serum. This increases T3 clearance and rate of peripheral deiodination (conversion of T4 to T3), which shifts the homeostasis of the pituitary–thyroid axis toward type 2 allostasis and ultimately leads to a higher basal metabolic rate;The findings on the association of the TT genotype of rs1800849 in the *UCP3* gene with increased FT3 levels in blood and with body weight deficiency demonstrate that the uncoupling protein UCP3 in skeletal muscle mitochondria actively participates in the processes of “uncoupling to survive”. This involves creating a proton leak in the respiratory chain, resulting in the conversion of electrical charge into thermal energy at the expense of ATP synthesis. A secondary effect of UCP3-dependent “uncoupling to survive” is likely to be competition with anabolic pathways for ATP, which may affect BSA, growth, and weight.

## Figures and Tables

**Figure 1 biology-14-00454-f001:**
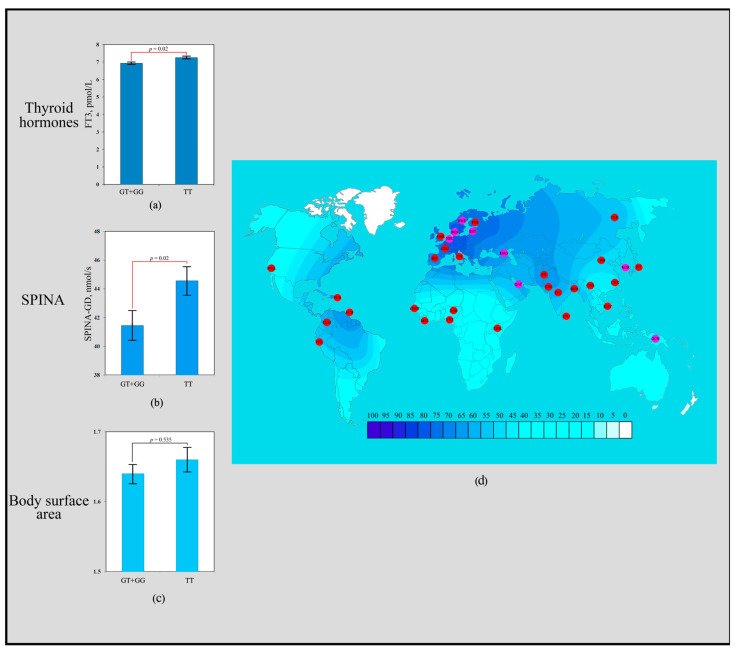
Effect of the rs3811787 of *UCP1* gene in human adaptation to cold: (**a**) The levels of FT3 as a function of rs3811787 genotypes of the *UCP1* gene; (**b**) Association analysis of the TT rs3811787 genotype between SPINA-GD; (**c**) Association analysis of the TT rs3811787 genotype between BSA; (**d**) The geographical distribution of the frequencies of the T rs3811787 (*UCP1*) allele in global human populations to identify signals of natural selection for cold climate adaptation.

**Figure 2 biology-14-00454-f002:**
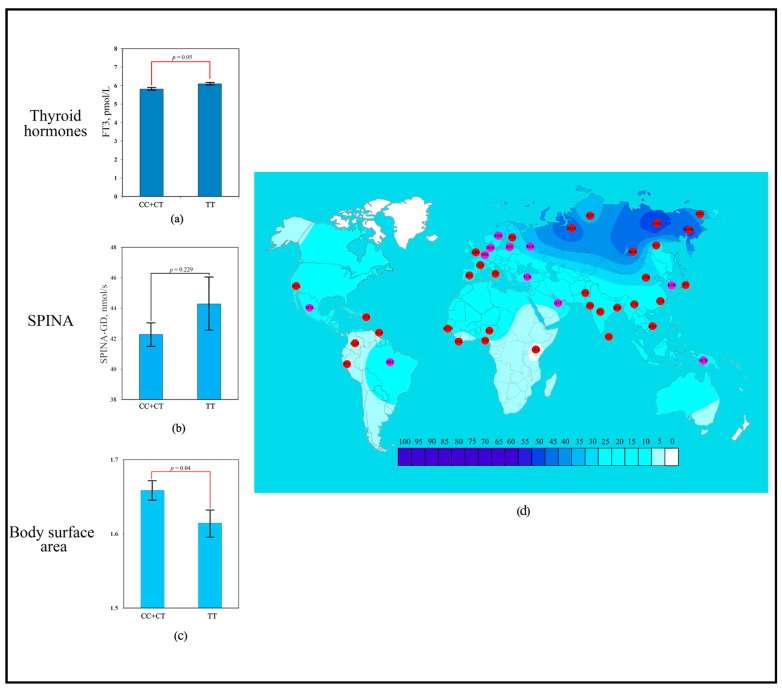
Effect of the rs1800849 of *UCP3* gene on human adaptation to cold: (**a**) The levels of FT3 as a function of rs1800849 genotypes of the *UCP3* gene; (**b**) Association analysis of TT rs1800849 genotype between SPINA-GD; (**c**) Association analysis of TT rs1800849 genotype between BSA; (**d**) The geographical distribution of frequencies of the T rs1800849 (*UCP3*) allele in global human populations to identify signals of natural selection for cold climate adaptation.

**Figure 3 biology-14-00454-f003:**
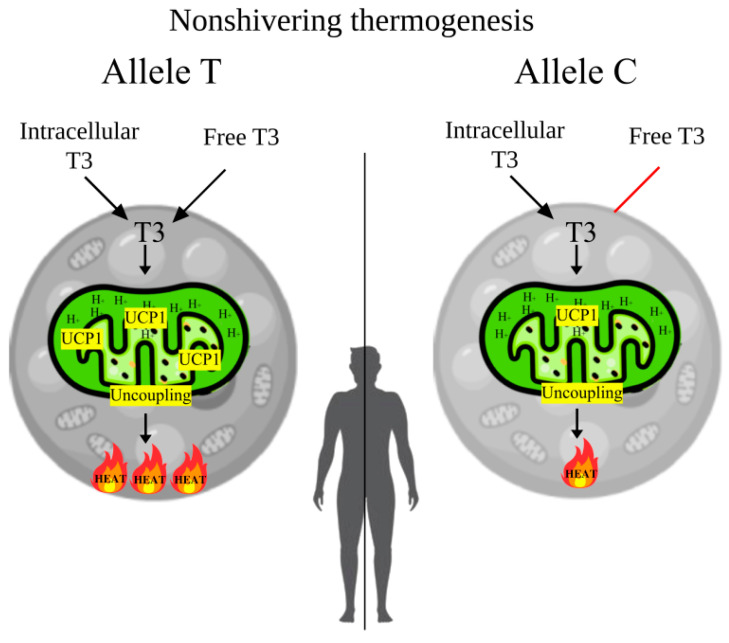
The mechanism of BAT-mediated regulation of thyroid hormones and basal metabolic rate from the allelic variants rs3811787 of the *UCP1* gene. **Note**. In the carriers of the T allele, with the active form of the UCP1 protein, brown adipocytes mitochondria use higher concentrations of T3 for increased heat generation during adaptive thermogenesis, which increases T3 clearance and in response increases peripheral deiodination (SPINA-GD), which raises blood levels of FT3 and basal metabolic rate. In carriers of the C allele, with a less active form of the protein, brown adipocytes will use less T3, which will not significantly increase SPINA-GD and blood levels of FT3, hence the metabolic rate will not significantly improve.

**Figure 4 biology-14-00454-f004:**
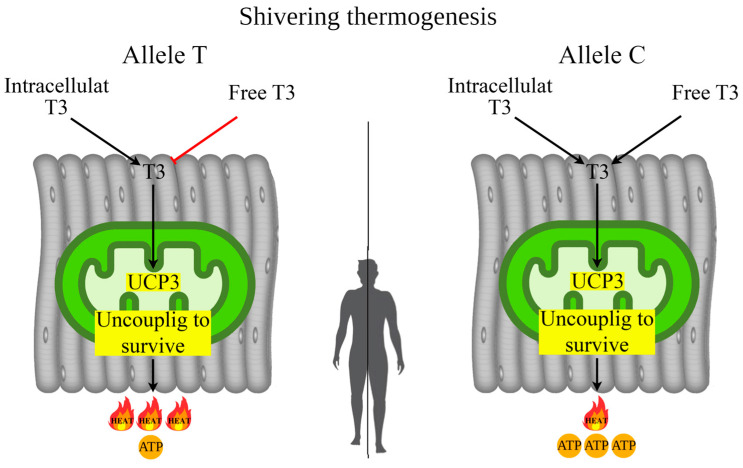
Possible mechanism of UCP3-dependent “uncoupling to survive” during shivering thermogenesis depending on allelic variants of rs1800849 of the *UCP3* gene in skeletal muscle. **Note.** In the carriers of the T allele (rs1800849) with increased UCP3 expression in skeletal muscle, heat generation during adaptive thermogenesis will occur mainly through “uncoupling to survive” without utilizing serum T3. In the carriers of the C allele (rs1800849) for enhanced UCP3 expression, skeletal muscle mitochondria will utilize T3 to enhance heat generation from “uncoupling to survive”, so serum T3 levels will be low compared to T allele carriers who do not utilize T3.

**Table 1 biology-14-00454-t001:** List of primer sequences, annealing temperature, and allelic profiles of the studied SNPs.

Gene	Chromosomal Region	SNP	Physical Location	Primer Sequence	Annealing Temperature/ Time, Product Size	Restriction Enzymes, Allele Sizes (bp)
UCP1	4q31.1	rs1800592 −3826A > G	the promoter region	F: 5′-ACATTTTGTGCAGCGATTTCTG-3′ R: 5′-TTCACCACTTCTGACAGGCT-3′	56 °C/45 s, 301 bp	Ksp22I A = 265 + 36 G = 301
		rs3811787 −412A > C	5′-flanking region	F: 5′-CCTTCTGTCACCCTTTGGCTGCACACCTTCGCC-3′ R: 5′-TGACAAGTTCAGAGTGCTCTT-3′	57 °C/45 s, 296 bp	Bst2UI T = 262 + 34 G = 296
UCP2	11q13.4	rs659366 −866G > A	5′-proximal region	F: 5′-AGCGTGACCTCACGCTCCTA-3′ R: 5′-GACTGAACGTCTTTGGGACTCCGT-3′	59 °C/45 s, 299 bp	BspFNI T = 178 + 121 C = 299
		rs660339 Ala55Val	exon 4	F: 5′-TTGACAGAATCATACAGGCCGA-3′ R: 5′-TTGGAGCATCGAGATGACTG-3′	53.8 °C/45 s, 392 bp	Bst4CI G = 392 A = 153 + 110
UCP3	11q13.4	rs1800849 −55C > T	the promoter region	F: 5′-CCTTGTCACCAAGGAAGCGTCCACAGCTT-3′ R: 5′-CTTCTGGCTTGGCACTGGTCTTATACACCC-3′	59 °C/45 s, 215 bp	SmaI C = 185 + 30 T = 215
		rs2075577 Tyr210Tyr	exon 3	F: 5′-GGGACTGGAACCCAAGTCT-3′ R: 5′-ACGACATCCTCAAGGAGAAGCTGCTGGAGTA-3′	58 °C/45 s, 249 bp	RsaNI G = 218 + 32 A = 249
UCP4	6p11.2-q12	rs9472817 C > G	intron 8	F: 5′-CCAAAGCCCTTGGCAATACC-3′ R: 5′-AACCTGCTGCACAGATTGTTGGGGAAATTCATC-3′	51.2 °C/45 s, 366 bp	Taq I C = 332 + 34 G = 366
UCP5	Xq24	rs1010978 T > C	intron 3	F: 5′-TTTGTATATGGCGGCCTTGC-3′ R: 5′-CTTAAGCCTAGCAAACTAACAAATCACTAAAGT-3′	59.5 °C/45 s, 284 bp	RsaNI T = 191 + 59 + 34 C = 191 + 93
UCP6	13q14.13	rs9526067 A > T	intron 5	F: 5′-CTCAGAGCCTCCAGTTTCCT-3′ R: 5′-GTTGTTGTTTGGTCTTTTGCTCTTTTTTTTTAT-3′	56.3 °C/45 s, 250 bp	SmiI T = 250 A = 211 + 39

**Table 2 biology-14-00454-t002:** Strengths of influence (in points) of four parameters on human adaptation to the cold.

Criteria	The Presence of Associations	High Statistical Significance *p* < 0.01	Average Statistical Significance *p* = 0.02–0.03		Low Statistical Significance *p* = 0.04–0.05
1	TSH	3 points	2 points		1 point
	FT3	3 points	2 points		1 point
	FT4	3 points	2 points		1 point
2	SPINA-GT	3 points	2 points		1 point
	SPINA-GD	3 points	2 points		1 point
3	BSA	3 points	2 points		1 point
4	**The presence of directional selection of natural signals for polymorphic variants of *UCP* genes**				
		Selection signals for Arctic and temperate climates		Selection signals for tropical and equatorial climate	
		3 points		0 points	

**Table 3 biology-14-00454-t003:** Assessment of the contribution of polymorphic variants of *UCP* genes to human adaptation to cold.

The Criterions	*UCP1* rs1800592	*UCP1* rs3811787	*UCP2* rs659366	*UCP2* rs660339	*UCP3* rs1800849	*UCP3* rs2075577	*UCP4* rs9472817	*UCP5* rs1010978	*UCP6* rs9526067
1 ^a^	0	2	0	0	2	0	0	2	2
2 ^a^	0	2	0	0	0	0	0	0	3
3 ^a^	0	0	0	0	1	0	0	0	0
4 ^b^	3	3	0	0	3	0	0	0	0
Total	3	7	0	0	6	0	0	2	5

^a^—the data obtained using association analysis; ^b^—the data obtained using geostatistical methods (spatial interpolation).

## Data Availability

The data presented in this study are available on request from the corresponding author.

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
