# Peer review of "The Evaluation of Significance of Uncoupling Protein Genes UCP1, UCP2, UCP3, UCP4, UCP5, and UCP6 in Human Adaptation to Cold Climates"

_biology, 2025, doi:10.3390/biology14050454_

Round 1
Reviewer 1 Report
Comments and Suggestions for Authors
The study does not provide new results that contribute to the problem. The presented analyses (methods) are not sufficient for publication in the journal in question.
1) The text of the article constantly talks about the role of a gene in cold adaptation. A gene codes for a protein, and proteins can influence physiological functions.
2) Do the authors present their own research results and an analysis of other authors' data? At a minimum, a study design is needed in the materials and methods. In the form presented, it is difficult to understand the research approach
3) Figures 3 and 4 are not based on the authors' own materials, and the discussion is highly speculative.
The article needs correction of the English language. I recommend that the authors contact a native speaker or use a special editorial service
Author Response
Response to Reviewer 1 Comments |
||
1. Summary |
|
|
We appreciate the time and efforts that the editor and the reviewers have dedicated to providing valuable feedback on our manuscript. All comments have been taken into account, and corresponding changes have been made directly to the manuscript where appropriate. We have highlighted the changes within the manuscript. The revisions have been approved by all authors. Here is a point-by-point response to the reviewers’ comments.
|
||
2. Questions for General Evaluation |
Reviewer’s Evaluation |
Response and Revisions |
Does the introduction provide sufficient background and include all relevant references? |
Yes |
|
Is the research design appropriate? |
Not applicable |
|
Are the methods adequately described? |
Not applicable |
|
Are the results clearly presented? |
Not applicable |
|
Are the conclusions supported by the results?
|
Not applicable |
|
3. Point-by-point response to Comments and Suggestions for Authors |
||
Comments 1: The text of the article constantly talks about the role of a gene in cold adaptation. A gene codes for a protein, and proteins can influence physiological functions.
|
||
Response 1: The aim of this study was to search for indications of involvement of polymorphic variants of UCP1-6 genes in human adaptation to cold climate in Yakut population living in the coldest region of Siberia (t°minimum -71.2°Ð¡). Our study based on phenotypic features founded in people living in cold climates: 1.Polar T3 syndrome (reduced levels of the thyroid hormone T3) [Read et al., 1988;1989; Levy et al., 2013; Leonard et al., 2014; Snodgrass et al., 2008; Nikanorova et al., 2023], 2. Thyroid homeostasis (allostasis 2 type, parameter SPINA) [Chatzitomaris et al., 2017; Nikanorova et al., 2023], 3. Body surface area [Smirnova, 1973; Klevtsova, Smirnova, 1973; Alekseeva, 1977; Deryabin, 1990] 4. Presence of signals of direct selection to cold climate for polymorphic variants of UCPs genes. Respectively, we discuss the role of the UCPs genes associated with these phenotypic traits in the context of human adaptation to cold climate.
|
||
Comments 2: Do the authors present their own research results and an analysis of other authors' data?
|
||
Response 2: To detect signals of natural selection for polymorphic variants of the UCP1-6 genes, we used allele frequencies data from the open database “1000 Genomes Project” [57] and studies published by Stepanov et al. [49], Schnor et al. [58], and Verdi et al. [59]. Based on these data, we constructed maps of allele frequency distribution across populations of North and South America, Eurasia, and Africa. The other results were obtained based on our own researches.
Comments 3: At a minimum, a study design is needed in the materials and methods. In the form presented, it is difficult to understand the research approach. Response 3: We added the study design in the Supplementary materials (Chapter 1, Figure S1).
Comments 4: Figures 3 and 4 are not based on the authors' own materials, and the discussion is highly speculative.
Response 4: The discussion of Figure 3 and 4 is based on the found associations, generally accepted mechanisms of nonshivering and shivering thermogenesis, and the functional role of UCP1 and UCP3 genes in thermogenesis, which include known data, described earlier (Klingenberg et al., 1999 [1], Virtanen et al., [5], Garlid et al., [17], Cannon et al., 2004 [18], Bianco et al., 1988 [19], Carvalho et al., 1991 [20], Rabelo et al., 1995 [21], Sentis et al., 2021 [22], Gong et al., 1997 [23], Masaki et al., 1997 [24], Barbe et al., 2001 [25], de Lange et al., 2001 [26]). In this study we present new data about associations of rs3811787 (UCP1) with blood FT3 levels and peripheral deiodination indices (SPINA-GD), and for rs1800849 (UCP3), associations were found with blood FT3 levels, BSA and signals of natural selection to cold climate. In Figures 3 and 4 we summarized our data with previous data obtained by Khaskin, 1984; Stock et al., 1989; Israel et al., 1989; Bell et al., 1992; Bazhenov, 1998; Skulachev, 1998; Meigal et al., 2002; Cannon et al., 2004; Haman et al., 2006; Nedergaard et al., 2007; Wijers et al., 2008; van Marken Lichtenbelt et al., 2009; Zingaretti et al., 2009; Cypess et al., 2009; Virtanen et al., 2009; Saito et al., 2009; Nedergaard et al., 2011; Rowland, 2015; Pant, 2016; Saltykova et al., 2018.
|
||
4. Response to Comments on the Quality of English Language |
||
Point 1: The article needs correction of the English language. I recommend that the authors contact a native speaker or use a special editorial service.
|
||
Response 1: |
||
We revised the English language through by text of the manuscript. |
||
|

Reviewer 2 Report
Comments and Suggestions for Authors
The manuscript "The evaluation of significance of uncoupling protein genes UCP1, UCP2, UCP3, UCP4, UCP5 and UCP6 in human adaptation to cold climates" by Nikanorova et al provides valuable insights into the role of UCPs in human adaptation to extreme cold environments, specifically in the Yakut population. The identification of UCP1 and UCP3 as key polymorphic variants contributing to cold adaptation is an important contribution to understanding thermogenesis mechanisms.
I recommend publication of this paper with a few minor issues being addressed.
Point 1: The discussion of UCP3’s thermogenic function could be expanded to strengthen the conclusion.
Point 2: Addressing limitations, such as sample size, population specificity would improve the study’s robustness.
Point 3: Future studies could consider functional validation experiments to further confirm the proposed mechanisms.
Comments on the Quality of English Language
The language is generally of a high standard. With a few minor corrections and rephrasings for clarity, the passage will be even more polished.
Author Response
Response to Reviewer 2 Comments
|
||
1. Summary |
|
|
Thank you very much for taking the time to review this manuscript. All comments have been taken into account, and corresponding changes have been made directly to the manuscript where appropriate. We have highlighted the changes within the manuscript. The revisions have been approved by all authors. Here is a point-by-point response to the reviewers’ comments.
|
||
2. Questions for General Evaluation |
Reviewer’s Evaluation |
Response and Revisions |
Does the introduction provide sufficient background and include all relevant references? |
Can be improved |
|
Is the research design appropriate? |
Can be improved |
|
Are the methods adequately described? |
Can be improved |
|
Are the results clearly presented? |
Yes |
|
Are the conclusions supported by the results? |
Yes |
|
3. Point-by-point response to Comments and Suggestions for Authors |
||
Comments 1: The discussion of UCP3’s thermogenic function could be expanded to strengthen the conclusion.
|
||
Response 1: We expanded the discussion of the thermogenic function of UCP3.
Сhange: Ln355-364 “…Studies on model animals and in vivo models have confirmed the uncoupling ability of UCP3 in skeletal muscle mitochondria [69-73]. Mice with a UCP3 gene knockout (UCP3 KO) exhibited enhanced mitochondrial respiration, indicating the protein’s proton transport activity [69, 70]. In UCP1 knockout mice, the loss of thermogenesis in brown adipocytes was compensated by increased contractile thermogenesis in muscles, accompanied by elevated UCP3 expression [71]. Another study on UCP3 KO mice confirmed that UCP3 is a critical mediator of physiological thermogenesis in skeletal muscles [72]. An in vivo study demonstrated that UCP3 overexpression in skeletal muscles reduces the ATP synthesis-to-mitochondrial oxidation ratio, suggesting uncoupling via UCP3 [73].”
|
||
Comments 2: Addressing limitations, such as sample size, population specificity would improve the study’s robustness.
|
||
Response 2: For this study we focused on a large random population sample of Yakuts which was collected to ensure more representative data. The Yakuts inhabit one of the coldest regions on Earth, with extreme winter air temperatures (the record low air temperature was -71.2°C). Consequently, gene variants that aid survival in such harsh cold climates may have been naturally selected in this population. This makes the Yakuts an excellent model for studying cold adaptation mechanisms. It is known that the thermoregulatory capacity of the body and thyroid gland activity may decline with age due to physiological changes. Therefore, the current sample was age-balanced (mean age 19.73±1.99 years).
Comments 3: Future studies could consider functional validation experiments to further confirm the proposed mechanisms.
Response 3: We believe that mechanisms of nonshivering and shivering thermogenesis that we proposed will contribute to future researches in this field. In particular, it would be interesting to look at serum FT3 levels after cold exposure, which increases the expression of UCP1 and UCP3. We hope that the proposed mechanisms will form the basis for further research.
|
||
4. Response to Comments on the Quality of English Language |
||
Point 1: The language is generally of a high standard. With a few minor corrections and rephrasings for clarity, the passage will be even more polished. |
||
Response 1: We revised the English language through by text of the manuscript. |

Reviewer 3 Report
Comments and Suggestions for Authors
The manuscript titled "The evaluation of significance of uncoupling protein genes UCP1, UCP2, UCP3, UCP4, UCP5, and UCP6 in human adaptation to cold climates" investigates the role of uncoupling proteins (UCPs) in thermogenesis and their potential contribution to human adaptation to colder environments. The study aims to establish a link between genetic variations in these proteins and metabolic efficiency in cold adaptation. The paper presents valuable insights into the genetic basis of thermoregulation, which could have implications for evolutionary biology and medical genetics. The paper's strengths include a well-defined hypothesis, a thorough literature review, and a logical structuring of the research problem.
Comment 1: The argument regarding cold adaptation lacks direct genetic evidence from population studies. More citations from genome-wide association studies (GWAS) would strengthen this claim.
Comment 2: Would adding a column that indicates whether the data were derived from experimental studies or computational predictions be helpful?
Comment 3: The review provides a good overview of UCP genes but does not extensively cover recent advances in functional genomics related to thermogenesis. Citing more recent studies, if possible, (within the last five years) would enhance its relevance.
Comment 4: Have you considered alternative explanations for thermoregulation, such as epigenetic modifications or environmental factors?
Comment 5: Were any bioinformatics tools employed to analyze the functional impact of the genetic variants? If so, which ones?
Comment 6: How did you control for potential confounding factors like diet, activity level, and other metabolic genes?
Comment 7: What are your findings' clinical or practical implications for understanding metabolic disorders or obesity?
Comment 8: Do you plan to extend this study by incorporating physiological measurements (e.g., brown adipose tissue activity) alongside genetic analysis? (If so, please mention it).
Comment 9: Could future studies focus on experimental validation, such as CRISPR-based gene knockouts, to confirm the functional impact of these variants?
Author Response
Response to Reviewer 3 Comments
|
||||||||||||||||||||||||||||||||||||||||||||||||||||||||||||||
1. Summary |
|
|
||||||||||||||||||||||||||||||||||||||||||||||||||||||||||||
Thank you very much for taking the time to review this manuscript. All comments have been taken into account, and corresponding changes have been made directly to the manuscript where appropriate. We have highlighted the changes within the manuscript. The revisions have been approved by all authors. Here is a point-by-point response to the reviewers’ comments.
|
||||||||||||||||||||||||||||||||||||||||||||||||||||||||||||||
2. Questions for General Evaluation |
Reviewer’s Evaluation |
Response and Revisions |
||||||||||||||||||||||||||||||||||||||||||||||||||||||||||||
Does the introduction provide sufficient background and include all relevant references? |
Yes |
|
||||||||||||||||||||||||||||||||||||||||||||||||||||||||||||
Is the research design appropriate? |
Yes |
|
||||||||||||||||||||||||||||||||||||||||||||||||||||||||||||
Are the methods adequately described? |
Yes |
|
||||||||||||||||||||||||||||||||||||||||||||||||||||||||||||
Are the results clearly presented? |
Yes |
|
||||||||||||||||||||||||||||||||||||||||||||||||||||||||||||
Are the conclusions supported by the results? |
Can be improved |
|
||||||||||||||||||||||||||||||||||||||||||||||||||||||||||||
3. Point-by-point response to Comments and Suggestions for Authors |
||||||||||||||||||||||||||||||||||||||||||||||||||||||||||||||
Comments 1: The argument regarding cold adaptation lacks direct genetic evidence from population studies. More citations from genome-wide association studies (GWAS) would strengthen this claim.
|
||||||||||||||||||||||||||||||||||||||||||||||||||||||||||||||
Response 1: We added reference to the recent study on cold adaptation in Upper Paleolithic hunter-gatherers of Eastern Eurasia (the Jomon people) that used the GWAS method [Watanabe et al., 2024].
Changes: Ln107-112 “…A study focused on identifying cold adaptation traits in ancient hunter-gatherers in Japan (the Jomon people) revealed associations between four SNPs in the UCP1 gene (rs3113195, rs12502572, rs1800592, rs4956451) and the nonshivering thermogenesis phenotype [50]. It has been hypothesized that East Eurasian hunter-gatherers adapted to cold climates through BAT thermogenesis mediated by UCP1 [50]…"
|
||||||||||||||||||||||||||||||||||||||||||||||||||||||||||||||
Comments 2: Would adding a column that indicates whether the data were derived from experimental studies or computational predictions be helpful?
|
||||||||||||||||||||||||||||||||||||||||||||||||||||||||||||||
Response 2: We have taken your comment into account and changed the table.
Changes: “Table 3. Assessment of the contribution of polymorphic variants of UCPs genes to human adaptation to cold.
a – the data obtained using association analysis; b – the data obtained using geostatistical methods (spatial interpolation).”
|
||||||||||||||||||||||||||||||||||||||||||||||||||||||||||||||
Comments 3: The review provides a good overview of UCP genes but does not extensively cover recent advances in functional genomics related to thermogenesis. Citing more recent studies, if possible, (within the last five years) would enhance its relevance.
|
||||||||||||||||||||||||||||||||||||||||||||||||||||||||||||||
Response 3: The functional genomics of UCP genes has already been extensively investigated in previous researches (Enerback et al., 1997; Gimeno et al., 1997; Boss et al., 1997; Samec et al., 1998; Sanchis et al., 1998; Mao et al., 1999; Vidal-Puig et al., 2000; Li et al., 2000; Reilly, Thompson, 2000; Gong et al., 2000; Ricquier et al., 2000; Chan et al., 2001; Yang et al., 2002; Echtay et al., 2002; Zackova et al., 2003; Liebig et al., 2004; Kontani et al., 2005; Haguenauer et al., 2005; Echtay et al., 2007; Shabalina et al., 2010; Ramsden et al., 2012; Hoang et al., 2015; Hilse et al., 2016; Riley et al., 2016; Macher et al., 2018; Gorgoglione et al., 2019). But we have not found recent studies within the last five years. However, we believe that our results provide a new relevance for further studies on the functional genomics of UCP genes associated with thermogenesis, especially on the topic of “uncoupling to survive” in skeletal muscles.
|
||||||||||||||||||||||||||||||||||||||||||||||||||||||||||||||
Comments 4: Have you considered alternative explanations for thermoregulation, such as epigenetic modifications or environmental factors?
|
||||||||||||||||||||||||||||||||||||||||||||||||||||||||||||||
Response 4: In the present study, associations were identified between SNPs of the UCP1 and UCP3 genes and phenotypic traits such as FT3, parameters of peripheral deiodination (SPINA-GD), and BSA. Our findings suggest that epigenetic modifications do not play a significant role in the observed associations, as the SNPs demonstrate significant links with the phenotypes. However, for other UCP gene SNPs that didn’t show associations with phenotypes, the influence of epigenetic mechanisms couldn’t be excluded. Especially for UCP2, UCP4 and UCP5 as they show selection signals for equatorial climates. When interpreting the results, we accounted for the effects of air temperature, based on previously published studies. Specifically, it has been shown that blood thyroid hormone levels may fluctuate in response to low air temperatures [32,34,37], linked to type 2 thyroid allostasis [33,37]. Another study indicated that these fluctuations are associated with BAT activity under cold air exposure [65]. Therefore, other environmental factors were not considered as alternative explanations for thermoregulation.
|
||||||||||||||||||||||||||||||||||||||||||||||||||||||||||||||
Comments 5: Were any bioinformatics tools employed to analyze the functional impact of the genetic variants? If so, which ones?
|
||||||||||||||||||||||||||||||||||||||||||||||||||||||||||||||
Response 5: We didn't use such tools.
|
||||||||||||||||||||||||||||||||||||||||||||||||||||||||||||||
Comments 6: How did you control for potential confounding factors like diet, activity level, and other metabolic genes?
|
||||||||||||||||||||||||||||||||||||||||||||||||||||||||||||||
Response 6: In our study, we did not control such factors as diet, physical activity level, and the influence of other metabolic genes. We did not take into account the influence of these factors, because such exceptions seems artificial in relation to the random sample of people living in natural conditions of extremely low temperature and excluding these factors can significantly affect the frequencies of the studied genes. We suppose that the random sample is the most representative for the search for human adaptations to cold climates.
|
||||||||||||||||||||||||||||||||||||||||||||||||||||||||||||||
Comments 7: What are your findings' clinical or practical implications for understanding metabolic disorders or obesity?
|
||||||||||||||||||||||||||||||||||||||||||||||||||||||||||||||
Response 7: Over the last decade, BAT has received much attention as a potential target for the treatment of obesity. The main interest has been related to nonshivering thermogenesis, which increases energy expenditure without physical activity. In addition, it has been found that white adipocytes can transdifferentiate into brown adipoites (browning), which may be useful for obesity therapy. In the present study, we show for the first time that the process of “uncoupling to survive” in skeletal muscles may be a more energy-consuming process directly affecting weight than non-shivering thermogenesis in BAT. We hope that our results on UCP3-dependent “uncoupling to survive” will be useful for future studies aimed at combating obesity.
|
||||||||||||||||||||||||||||||||||||||||||||||||||||||||||||||
Comments 8: Do you plan to extend this study by incorporating physiological measurements (e.g., brown adipose tissue activity) alongside genetic analysis? (If so, please mention it).
|
||||||||||||||||||||||||||||||||||||||||||||||||||||||||||||||
Response 8: Currently we do not plan to expand this study.
|
||||||||||||||||||||||||||||||||||||||||||||||||||||||||||||||
Comments 9: Could future studies focus on experimental validation, such as CRISPR-based gene knockouts, to confirm the functional impact of these variants?
|
||||||||||||||||||||||||||||||||||||||||||||||||||||||||||||||
Response 9: The functional genomics of UCP genes has already been extensively investigated in previous researches (Enerback et al., 1997; Gimeno et al., 1997; Boss et al., 1997; Samec et al., 1998; Sanchis et al., 1998; Mao et al., 1999; Vidal-Puig et al., 2000; Li et al., 2000; Reilly, Thompson, 2000; Gong et al., 2000; Ricquier et al., 2000; Chan et al., 2001; Yang et al., 2002; Echtay et al., 2002; Zackova et al., 2003; Liebig et al., 2004; Kontani et al., 2005; Haguenauer et al., 2005; Echtay et al., 2007; Shabalina et al., 2010; Ramsden et al., 2012; Hoang et al., 2015; Hilse et al., 2016; Riley et al., 2016; Macher et al., 2018; Gorgoglione et al., 2019). We believe that the current use of CRISPR technology will most likely be aimed at treating monogenic forms of obesity and metabolic disorders. |

Round 2
Reviewer 1 Report
Comments and Suggestions for Authors
I understand the authors' explanations and goals, which are stated in the responses to most of my comments. However, no actual results using additional methods are presented.
Author Response
No comments
Reviewer 3 Report
Comments and Suggestions for Authors
All comments were addressed. Thank you!
Author Response
Thank you very much for you kind review!